# Propositional Inference for IoT Based Dosage Calibration System Using Private Patient-Specific Prescription against Fatal Dosages

**DOI:** 10.3390/s23010336

**Published:** 2022-12-28

**Authors:** Karthikeyan Gopalakrishnan, Arunkumar Balakrishnan, Kousalya Govardhanan, Sadagopan Selvarasu

**Affiliations:** 1Department of Computer Science and Engineering, Coimbatore Institute of Technology, Coimbatore 641014, India; 2School of Computer Science and Engineering, VIT-AP University, Amaravati 522237, India; 3School of Computing, SRM Institute of Science and Technology, Kattankulathur 603203, India

**Keywords:** internet of things (IoT), blockchain, basal, bolus, state change, immutability

## Abstract

IoT-based insulin pumps are used to deliver precise quantities of insulin to diabetic patients to regulate blood glucose levels. Generally, these levels correspond to the dietary patterns observed at time intervals that vary between patients. However, any misrepresentation in insulin levels may lead to fatal consequences. As a result, most IoT-based insulin pumps are rejected due to the possibility of external threats, which include software and hardware attacks. However, IoT-based insulin pumps are extremely useful in real-time patient monitoring, and for controlled insulin delivery to the patient based on their current glucose level. We propose a blockchain-based method to protect against the above-mentioned attacks. The system creates a patient-specific private blockchain wherein the dosage information is added as a new block by obtaining the approval of the doctor, chief doctor, nurse, and caretaker of the patient who are authorized blockchain miners. Secondly, it securely transfers prescription data, such as dosage quantity and time of delivery, to the IoT insulin pump, which ensures the dosage information is not modified during transit before insulin administration to the patient. The proposed approach uses a state-behavior-based solution that detects anomalies in the behavior of the insulin pump via temporal data analysis and immutable ledger verification, which are designed to eliminate fatal dosages in case of anomalies. The system is designed to work within binary outcome conditions, i.e., it verifies and delivers dosage or halts. There is no middle ground that an attacker can exploit, resulting in accountability for the system.

## 1. Introduction

An insulin pump is a medical device designed to deliver insulin to patients suffering from Type 1 or Type 2 diabetes mellitus [1]. The system is designed to deliver insulin based on a logic specifically tailored to the patients’ degree of ailment. The outcome of these devices eventually leads to a balance in the blood sugar present in the individual [2]. Most of these devices are based on a preset program fed into the system based on manual measurements of blood sugar done by a separate system. The adjustments are made manually or via wireless access mechanisms. Additionally, the patient is expected to wear the device on their body and has limited knowledge to decide whether the pump is functioning appropriately or not. In other words, the proper functioning of the insulin pump can be verified by any of the following ways: (1) the person’s periodic glucose measurements attest the pumps’ integrity as to whether it is properly working or not; (2) if the person is able to function in a relatively normal fashion without succumbing to fatigue, loss of consciousness or fatality. The primary reason for this is that a pump is an actuator controlled by a processor without an active sensor justifying the processors’ actions. An IoT enhanced Insulin pump has been introduced that can directly communicate with a node (Laptop, Smart Phone) via the internet for a range of purposes including, but not limited to, data collection, program adjustment, device monitoring and patient status. The device can be monitored in real-time, which results in better care offered to patients that includes fatal contingencies. However, the IoT enhancement was rendered unusable due to security concerns. The primary concern is that it is open to circumvention by skilled hackers, which forces the system to work beyond the norms stipulated by the care giver [3,4,5].


*“Manufacturers of medical IoT devices should be prioritizing security, especially considering the potential detrimental consequences of a breach.”*
(Catherine Norcom, Hardware hacker for IBM’s X-Force Red)

The security risks in an IoT insulin pump have significant consequences ranging from a loss of Quality of Life (QoL) to severe outcomes such as fatality, e.g., untreated hypoglycemia resulting in coma or death. If such negligence is initiated via a hack, then it is equivalent to murder, thereby turning a life-saving medical device into a lethal weapon. The outcome of an unauthorized access can be summarized in the following points:The dosage is increased resulting in extreme decomposition of blood glucose. The nominal result is fatigue, and in extreme cases, fatality.The dosage is decreased resulting in build-up of glucose, leading to hyperglycemia. The nominal result is shortness of breath and nausea, and in extreme cases, cardiovascular problems.The dosage dispersal frequency becomes chaotic. The basal and bolus factor is affected. The nominal result varies from patient to patient based on the severity of the disease.

Other complications include insulin variations causing the patient to become irresponsive to insulin, both naturally secreted or artificially introduced, thus causing further complications. In addition to medical reasons, the data related to the patient may be maliciously obtained for illicit study or personal slander of the patient. This is a critical factor that has resulted in negative opinions from high profile patients who have opted against IoT-enhanced insulin pumps, and who view the pump as a liability rather than an asset. 

In this paper, an attempt to integrate an insulin pump with an IoT processor was attempted. The countermeasures for IoT vulnerabilities were handled by private blockchain technology coupled with behavior analysis of dosage changes. First, we established a private blockchain to store dosage information that needs to be acknowledged by the doctor, chief doctor, nurse or a caretaker of the patient, who all act as miners for the patient-specific blockchain. Second, the dosage quantity and time of delivery is delivered securely to the IoT insulin pump as a regular blockchain block. Third, dosage quantity and duration cannot be modified during transmission before actually injecting insulin into the patient’s body.

The paper is divided into six sections. Section 2 highlights the background knowledge necessary to understand the workings of IoT-enabled medical actuators, using an insulin pump as an example. Closely associated literature is considered in Section 3. Section 4 discusses the role of the blockchain in securing against IoT vulnerabilities. Section 5 describes the mathematical proofs that corroborate the propositions made in the earlier sections. Section 6 concludes the research work and consolidates the contributions of proposed solution towards medical IoT devices.

## 2. Background and Challenges

This section first outlines the operations of a conventional insulin pump together with its limitations. Later, it highlights the elements of the proposed IoT-based insulin pump together with its possible attack vectors.

Conventional Insulin Pump. An insulin pump contains a processor that monitors an actuator mechanism that delivers insulin to the bloodstream via a catheter. The delivery mechanism has two different modes, namely, a basal rate and a bolus rate to deliver light and heavy doses, respectively, as prescribed by the medical practitioner.

All insulin pumps must consider four working conditions for nominal operations, and failure to do so renders the design useless. The situations to be addressed are outlined in Table 1.

Basal—Nominal amount of insulin injected periodically to maintain blood sugar. Period depends upon program.Bolus—Heavy dosage of insulin to be initiated by the patient at mealtime, or hyperglycemia symptoms affect quality of life.

The limitation is that the pump needs human intervention for operation at mealtimes and the associated rise in blood sugar. IoT-based pumps are limited by the same factor as there are no sensors to initiate operations in such circumstances.

An IoT insulin Pump must have the following components [6,7]:A sensor and actuator that collect data and effect quantifiable change. In the case of an insulin pump, the motor, piston and cannula together form the insulin delivery actuator.A processor with the insulin delivery program installed, which is subject to change based on human intervention through a communication module.A battery power source that runs the device.A communication module, which, in IoT pumps may represent a wired or wireless communication link. Wireless technologies include BT-WiFi, BLE (Bluetooth Low Energy) & ZigBee. The primary vulnerabilities recognized in IoT insulin pumps that resulted in recall were present in the communications module that negatively affected the processor module. The processor module, in turn, affects the rest of the IoT components in a cascading fashion.

Attacks that can be deployed against the IoT Insulin pump can be via hardware, i.e., physical sabotage of the device, or via software wherein the loaded program and communication ports are targeted to operate beyond intended parameters. Nearly all malfunctions can lead to fatalities with resulting loss of trust in the system. The primary factors affecting integrity are software-oriented attacks that allow malicious attacks.

## 3. Related Work

We aimed to augment a medical actuator that runs on preset temporal programs with IoT (Internet of Things) technology. There have been many attempts at integration but real-time deployment is not recommended by security experts, as remote activation of a medical actuator is open to misuse by external agents with malicious agendas. Combining IoT with any medical actuator makes the system a possible security risk [8,9,10], because vulnerabilities of the IoT system, such as authentication failure or loss of integrity, will threaten the life of the patient. Since the potential outcomes include fatality, there is strict opposition against IoT/ medical actuator integration. The other concern is, since there is a possibility of system failure due to external agents, there is potential for the medical device to become a potential weapon [11,12,13]. Attacker from a remote location could cause the actuator to execute a malicious code that alters system functionality. The probability of a fatal outcome is high, as most medical actuators are deployed for patients suffering from a chronic form of the disease that the medical device treats. Some case studies are outlined below.

### 3.1. A Brief Chronology of Medical Device Security—A.J Burns et al. [14]

Theis paper outlined the threats posed by IoT device integration into medical instruments in four timeline periods. These periods evaluated IoT devices from the perspectives of accidental failures to intentional threats, stating that IoT devices may harm patients to the point of fatality if measures are not put in place. The paper took into consideration devices that treat arrhythmia (i.e., a pacemaker), and studies that involved insulin dosage failures. The issues of software update, accidental failure and intentional hacking were highlighted as the primary failure of software-based medical instruments. The paper recommended the following steps to ensure safe usage. They are [14]

Identification. Identify processes and assets needing protection,Protection. Define available safeguards.Detection. Devise incident detection techniques.Response. Formulate a response plan.Recovery. Formalize a recovery plan.

These recommendations were successfully observed in the system proposed in the paper. In addition, the transparency of blockchain made the system open to only legitimate changes.

### 3.2. Deep Insecurities: The Internet of Things Shifts Technology Risk—Samuel Greengard [15]

This paper focused entirely on the dangers of integrating medical instruments with programming. It states that programming is a closed-system-based environment that has yet to transcend the drawbacks that arise due to connectivity. The paper also considered the cases of pacemaker and insulin delivery devices as case study examples. The paper insisted that programming-based medical instruments are susceptible to external compromise. Therefore, they recommended security systems to address failures, and threat mitigation systems to be put into place to establish trust in programmable medical devices. 

The paper also recommended a transparent system with explicit rules and guidelines that could be seen by authorized entities to verify conformance to protocols. The transparency recommendation serves additionally for legal institutions to verify and, in the case of failure, exact justice for critical stakeholders, i.e., the patients who stand to lose their lives.

The paper expresses concern over the lack of guidelines in existing embedded medical systems. It emphasizes that the lack of trust is a direct reflection of the absence of security protocols and guidelines. It recommends that a reward/punishment-based organization of programmable medical devices be established to ensure conformance.

The proposed system in this paper is suited for transparency-based operation of insulin delivery where all data is passed through a blockchain for conformance and security. All data are open to authorized personnel; thus, if a future reward-based environment is needed, the proposed system can accommodate readily. 

### 3.3. Internet of Things (IoT): Application Systems and Security Vulnerabilities [8]

This work states that any system with characteristics of remote management, dynamic topology, resource constraints and wireless communication medium is exposed and can be exploited for illegitimate purposes. It states that “the smarter the device the more exposed it will be”. Smart homes, wearable devices, smart cities, smart health, and transportation and traffic management were considered. [16,17,18,19,20].

From study it was observed that attack vectors (surfaces) are:


The IoT Device memoryIoT Web InterfacesIoT device network servicesIoT Device Cloud connectivity

IoT device-to-device connectivityIoT AAA ServicesIoT data storage methodsIoT device software updates


The paper emphasized that all the above attack surfaces are susceptible to ten different types of attacks, all of which could cripple the existing system completely [8]. Additionally, study of the current lack of hardware and software support in IoT was prioritized. It was inferred that a massive security threat is present on IoT devices and on networks connected to them [21].

Propagation of the same threats expose medical IoT devices to attack, which in turn poses a threat to patient well being. The solution proposed directly annuls all the threats mentioned above, providing total security to medical IoT devices.

### 3.4. Security and Privacy Considerations for IoT Application on Smart Grids: Survey and Research Challenges [22,23]

The paper addresses incorporation IoT applications to smart grids. A smart grid [23] is a cluster of micro-grids with multiple layers that provides distribution of power with a “collective intelligence”, facilitating bidirectional communication of power and, specifically, data. The paper also recognizes the advantage of 6LoWPAN over standard IPv6 packets as a suitable technology for resource constrained devices. The detrimental factors of IoT identified are stated below.


InteroperabilityCreation of hybrid structuresCustomer behavior variations

Data leaksNew data AggregationManagement of vast amounts of data.


The primary challenge in IoT is security and privacy of smart meters (IoT devices) is acknowledged, and it is emphasized that IoT is an intersection of multiple technologies not limited only to information technology; thus any impact on it propagates to many layers continually. In addition, the paper iterates the need for low overhead in both communication and computation.

It can be inferred from the paper that anomalies and data leaks in transit can cause significant loss of integrity. This loss in the context of medical IoT devices can lead to catastrophic consequences. Thus, the proposed model invokes a two-phase verification system to counter this.

### 3.5. The Security Challenges in the IoT Enabled Cyber-Physical Systems and Opportunities for Evolutionary Computing and Other Computational Intelligence [11]

This provide a general outlook on the many applications of IoT and, through it, the achievability of a “Smart-Factory” and its industrial and domestic variations. The work recognizes IoT’s capacities and its deficiencies in terms of security, and expresses the need for two factors to enhance IoT: (1) security architecture, and (2) security by design. 

The paper states that cyber security algorithms should be deployed at the architectural level, the data level and the data mining level to ensure privacy and prevent data theft. At the architectural level, focusing on edge devices, the core network and cloud are equally important. The data must be authenticated at every level and analyzed.

In reference to the above recommendation, the proposed system is based on private blockchain architecture to ensure that the flow of data is within the safety parameters of logic.

### 3.6. IoT Security Framework for Building Trustworthy Smart Car Services [24]

The work describes IoT architecture called the IoT Security Development Framework (ISDF) for CAN (Car Area Network) which segregates the entire unit into four layers: (1) an end-devices layer; (2) a communications layer;(3) a services layer, and (4) an application layer. These layers are implemented in four planes to identify threat and threat mitigation solutions. The integrated system is called the VIMP (Vehicle Information and Management Portal).

It is proposed here that security in IoT should be addressed systemically, and not as an afterthought. IoT extends traditional networks, and all nodes in IoT communicate. A compromise propagates to both IoT and the traditional network [25]. It is also stated that traditional IT security protocols not feasible for IoT [25,26]. Additionally, computing capacities are limited in IoT, and the increase in IoT nodes increases the entry points for threatening entities. An absence of systems that identify, rank, strategize and mitigate threats is observed. The need for an intrusion detection system (IDS) that detects, identifies, strategizes and mitigates threat is stated [27,28]. A compromise in IoT affects three areas in all four layers of the VIMP: (1) control of the system; (2) human life safety, and (3) time.

The replacement of “after-thought” security measures for IoT is incorporated into the proposed system, where all data are open in an immutable blockchain architecture. The intrusion detection system is accomplished by a two-phase verification mentioned below. 

### 3.7. Summary of Inferences

Comprehensively, the key point addressed all literature is that the IoT is insecure, and any purpose served through it has a possibility of being compromised, and the nodes themselves may become an exploitable attack surface. The advantages and risks associated with IoT are shown in Table 2.

## 4. Proposed Secure IoT Based Insulin Dosage Dispensary System Using Patient-Specific Private Blockchain

The key selling point of Blockchain technology is that it provides immutable accountability as a part of its storage and transmission protocols. Although it provides additional features, such as decentralized data storage and end-to-end encryption, its primary feature is that it supports integrity and availability through immutable redundancy [26,29]. Immutability is achieved through irreversible hash values that link one block to another. Redundancy exists so that if a corruption factor comes into play the uncorrupted nodes render it void. Presently, to corrupt an existing block chain, 51% of all the miners must act together to achieve this. This is virtually impossible as the identities of the miners themselves remain masked by the blockchain protocol. Secondly, the cost of breaking the blockchain is less rewarding than actually being a legitimate part of the distributed ledger network. Thus, the point of trust being established by the blockchain was used for our research work, and the solution for an IoT insulin pump is proposed as follows:

The proposed system consists of an IoT-based insulin pump and a patient-specific private blockchain. The proposed IoT-based insulin pump and construction of patient-specific private blockchain approach establishes a patient-specific private blockchain wherein the members need to be authenticated, thereby limiting the participation of unknown persons as a part of the blockchain with ability to add and validate the blocks in the Blockchain. Here, the proposed approach designates the doctor, chief doctor, nurse and caretaker of the patient as miners. The patient-specific doctor is authorized to add blocks while the collective acknowledgement of chief doctor, nurse and care-taker validate the addition of new blocks into the blockchain. In short, for every patient there is a separate, private blockchain created and unknown user participation is prohibited. 

The limitations of the existing paradigm are that each patient can only have one new chain created for research. A blank header that separates patients with a hash identification is taken into consideration when adding data from multiple patients in a single chain. One chain holds all the patient data; however, it is restricted because it is an open data chain between credentialed medical professionals. For security reasons, it is advised that only one chain per authorized group of medical practitioners maintains their patients’ information. 

At this point, each block contains dosage information such as dosage quantity and time of delivery. The instruction contained within the IoT pump would be the latest block containing the most recent prescription for the patient. A new block added to the blockchain means the dosage information has been modified and updated to the blockchain. In the blockchain, addition of a new block initiates an event which will be notified to the IoT-based insulin pump. Then, the pump reads the contents of the latest block and verifies whether the previous hash value of the new block matches with the present block’s current hash value [28]. 

In Bitcoin contexts, a block typically has a size of 1 MB and is constrained by a 4-byte field that limits the amount of data that may be stored in a block, whereas the current chain is required to store insulin prescription exclusively. Thus, reducing the minimum size of a block is not mandatory, as the private Blockchain only communicates the dose information within the new prescription that is contained in the most recent block and not the entire chain, together with the hash content for IoT-pump validation. For the experiment, a 2 GB Micro SD/memory card and a 32 kB internal flash memory delivered the enhanced IoT-pump its optimum performance. 

A successful hash match implies that the data have come from a legitimate source. The pump then extracts the recent dosage information and applies it to the insulin pump to deliver insulin to patient’s body. The block diagram in Figure 1 depicts the proposed solution. The new dosage information block replaces the old block in the IoT pump. Each time, the IoT-based insulin pump conducts this step before actually injecting the insulin. This is to ensure that the device always uses the most recent dosage information, which is in congruence with the most recent prognosis made by physician. This ensures the operational integrity of the IoT pump. Hence, any attacker who attacks the IoT-based insulin pump directly cannot modify the dosage information since it is available in the blockchain. 

Any attacker who tries to modify the dosage information in the blockchain needs to first compromise the authentication mechanism and masquerade as a member of that private blockchain. This is impossible because the miners are declared openly, authenticated, and fixed per patient as a clause of a smart contract. Thus, getting consensus from the doctor, chief doctor, nurse and caretaker to add a new block into the blockchain is mitigated by the inherent design of the private blockchain smart contract. Therefore, there is no means for an attacker to see the contents of the private blockchain unless they become a member through legitimate channels and a screening process, ensuring patient safety.

Another potential attack point occurs when the blockchain and IoT insulin pump communicate dosage information. This can be avoided by providing the block hash, which is encrypted using the symmetric AES technique to increase security, independent of the block’s content. To reduce processor burden, a key size of 128 bits was used. The process of introducing decryption required augmentation of an Arduino processor with a Raspberry Pi Pico/RP2040 with an 8 MB FLASH 264 KB SRAM. This reduces the chance of listening attacks occurring while data are being transmitted. Despite adding to the stress on IoT processors, the decryption process serves as a firewall against passive attacks.

The two data streams have a temporal difference during transmission, increasing the difficulty of a hack, thus providing two-fold protection to the block. If an attacker modifies the dosage information during transit, it affects the root of the Merkle tree which, in turn, affects the hash of that block. Thus, the IoT-based insulin pump receives the hash of the block contents it receives and compares it against the block hash which is received separately to ensure that no modification happened during the transit before actually using it for injecting insulin into the patient’s body.

Although all blocks in the chain are available to the designated miners, only the data in the most recent block gets transferred. These security safeguards guarantee that the IoT actuator follows the approved dose recommendations.

In short, the use of a patient-specific private blockchain eliminates the possibility of attack in the blockchain IoT-based insulin pump and during communication. Apart from this, the patient-specific nature of the blockchain ensures privacy and personalized security for the patient. 

The concise sequence of operations in the proposed system is as follows.

i.The patient-specific private blockchain is bootstrapped by its miners (i.e., doctor, chief doctor and caretaker of the patient)ii.The doctor decides the dosage information (i.e., specifies the quantity of insulin and the time of delivery) for the patient and generates the block in the chain containing the latest prescription information. The block is validated by the consensus mechanism that includes chief doctor, nurse and caretaker of the patient, and is appended to the blockchain.iii.The new block in the Blockchain initiates two different transactions to the pump [29,30,31]:It sends the block hash value in one transaction.The entire block (i.e., dosage information and hash value) is transmitted in another transaction.iv.The pump hashes the block contents and compares them against the block hash obtained in the separate transaction.If equal, go to step (v).Else, reject the new block and terminate the dosage procedure until a valid block arrives.v.The pump compares the previous hash of the new block with the current hash of the existing block [32].If equal, initiate insulin delivery.Else, reject the new block and terminate dosage until a valid block arrives.

The sequence of execution of the algorithm is represented in Figure 2.

From the sequence diagram, a functional partition of the diabetic patient from the rest of the system can be observed. This is to ensure that the blockchain-IoT solution will not be a source of problems for the patient [33,34]. As outlined earlier any vulnerability has the possibility of fatality. It is important to remember that the cost of upgrading an insulin pump to include an Arduino-IoT control system is about $100 including software connectivity. The device itself costs about $1000; therefore, the cost of augmentation is roughly a one-tenth of the equipment’s initial cost, making it a relatively affordable investment in instrument safety. Our prototype in a closed-environment with simulated conditions, generated a proper response for a new prescription with a retrofitted blockchain contract with a delay of 10 min from the generation of test block. The delay was due to transmission and processing by undedicated servers (i3 with 2 GB RAM) and the inherent deficit in the IoT processor capacity.

The working of the proposed system and its security capabilities are demonstrated in the following case scenario.

Case Scenario: Out-of-context behavior can be used to trigger alerts in the system console as well as with all the healthcare providers. Sample behavior and violation states for the forenoon session of a diabetic patient employing an IoT insulin pump is shown in Table 3. From the table it can be inferred that whenever there is an inversion in the expected status, and the ongoing conditions are inverted, the situation demands a system review. The most critical context arises when inversion occurs without human authorization. Dosage switching from one state to another demands a dependable tracking mechanism. This is needed, as the change is backed by an incorruptible source. 

Solution: A blockchain can be employed as a solution to this problem. A blockchain thorough its hash chain mechanism and distributed ledgers ensures immutability to the data encompassed in it. This feature can be used to track state change in the system as well as track of out-of-context behavior. The data to be stored consist of the state table shown in Table 3. Instead of just triggering alerts to the care-giver, a copy of the situation is uploaded onto the blockchain. This ensures that, not only a trigger indicating a context switch occurred, but there is a permanent copy of the event that can be studied and mitigated. A state transition diagram for the state table is shown in Figure 3.

The above diagram clearly outlines the importance of the blockchain, i.e., all insulin pumps are designed to run perpetually. The only legitimate reasons for an insulin pump to terminate functionality are:Battery replacementInsulin cartridge replacementPatient opts to remove it for a period of time.

Thus, if the system encounters any other situation, it indicates chaotic behavior that must be studied. It can be inferred from the state diagram that chaotic behavior is equivalent to malicious behavior. The blockchain records this behavior and broadcasts it to the distributed clients of the chain. The chain may include the patient, health care provider and the developer. Each of these entities may be able to respond with a long-term solution resulting in minimal service interruption. Additionally, if out-of-context behavior is fed to the blockchain, the data generated by the IoT insulin pump contains a repository of all failure cases. These failure cases can only be accessed by legitimate users of the blockchain because of private key cryptographic algorithms. Thus, the problem, and any solutions derived, are abstracted from all illegitimate entities, resulting in complete accountability of the system.

## 5. Proof of Impenetrability through Propositional Calculus

In this section, an attempt to prove a causal relationship between the presence of malicious actions and insulin pump failures (at different states of operation) is outlined. The proof outlined uses temporal variable as the fundamental factor to detect anomalies. Subsequently, the case scenario solution described above is injected into a mathematical model to test the viability of the solution. Propositional calculus is employed to mathematically prove the causal relationship between the system and its outcomes is shown in Table 4. 

Propositional calculus is first used to establish the indisputable facts about the system [32]. In succession, the immutable facts are introduced to malicious states as axioms. The purpose is to prove that the system fails when expected and actual states mismatch. This is achieved by proving that the natural order axioms contradict the malicious axioms. Second, solution axioms are introduced that negate the contradiction introduced by malicious axioms, proving the annulment of infection. Logically, the original fully functioning system can be represented as a tautology, i.e., it remains “true” for all proper inputs. Malicious entities deviate from the protocols by introducing codes that fork the system’s operational parameters. The deviation results in “contradiction,” which is the mathematical equivalent of system collapse. The solution re-achieves tautology.

Scenario 1: System under attack. A system under attack will have mismatching states of operation. The mismatch will be incongruent with the time factor. The time factor is the basis of all operations in an insulin actuator. Thus, proper detection of interference will trigger the breaking of the insulin dosage loop. Secondly, it will also trigger an entry into the data register.

Axiom representation:Successful State: LFailure State: BC^H (or) !L

From the above inferences it is mathematically proven that the logic deployed in the insulin pump system completely discriminates a legitimate system from a corrupted system is shown in Table 5. This mutual exclusion ensures that the system either works nominally or halts. Halting ensures that the life of the patient is not compromised by the system breach.

## 6. Conclusions

An IoT insulin pump is a medical actuator that functions with temporal data-based programs. Public interest declined in such devices due to the risk posed by IoT-based actuators that were susceptible to external corruption. The proposed system is a reactive, behavior-based solution that has a binary response. Either it performs nominally or terminates, removing the fatal component of the IoT actuator. Private blockchain technology is employed as a middleman/ ledger to verify data handled by the IoT pump to ensure integrity of system execution. The system is based on logic that is transparent to all parties in question, making the system safe and accountable. These modifications ensure integrity of medical IoT devices, and establish credibility among people and medical practitioners, thus improving medical quality. Future scope of applications includes retrofitting similar pumps to automate intravenous drug delivery mechanisms, and to ensure accountability in all aspects of medical intervention.

## Figures and Tables

**Figure 1 sensors-23-00336-f001:**
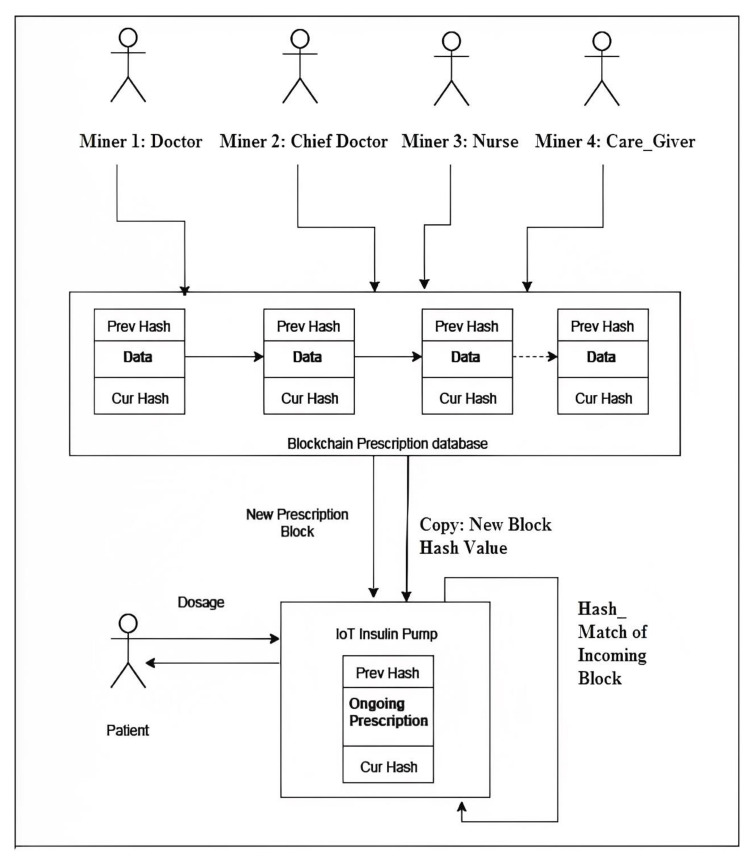
Block Diagram of a Blockchain-based IoT insulin pump [27].

**Figure 2 sensors-23-00336-f002:**
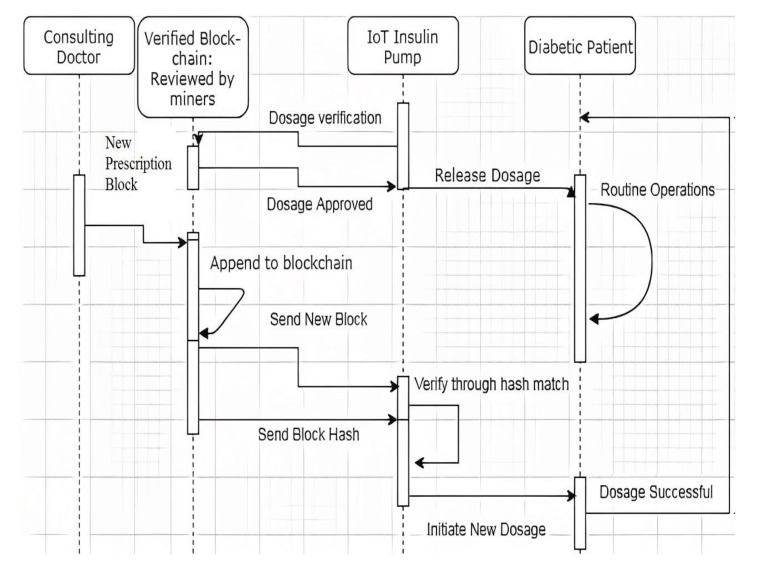
Working sequence of the Blockchain IoT Pump.

**Figure 3 sensors-23-00336-f003:**
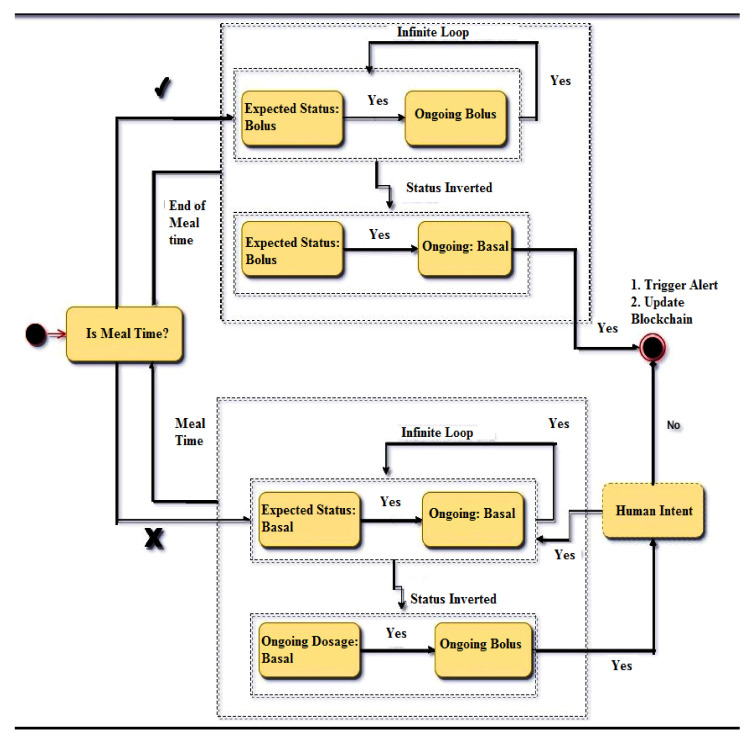
State transition diagram for an IoT/BC insulin pump [16].

**Table 1 sensors-23-00336-t001:** Scenario of patient status & response.

Meal Time	B. Sugar Threshold	Outcome	Duration	Programmable Factor
✓	✓	Bolus	1–4 h	Quantity of Insulin
✘	✓	Bolus	Symptoms abation	Quantity of Insulin
✘	✘	Basal	Periodic	Quantity of Insulin & Frequency
✓	✘	-	Patient recovery imminent	Abstinence

**Table 2 sensors-23-00336-t002:** Summary of the research pertaining to IoT threats and vulnerabilities.

Paper Name	Issues Addressed	Solution Proposed	Observations	IoT Relevance
A brief chronology of medical device security	Accidental threat. Intentional attack.	Five step security protocols.	Organized criteria for medical IoT.	Prescription of organization adopted.
Deep insecurities: the internet of things shifts technology risk	Closed system Programming. Lack of explicitness. Lack of guidelines.	NA	Recommendation of blockchain architecture resolution.	Need for external verifications with supervision.
Internet of Things (IoT): Application Systems and Security Vulnerabilities	Security Threat propagation Hardware Inadequacy	None Provides consolidation of threats only	IoT: An effective tool. Most vulnerable. Difficult to defend.	Vulnerability implies fatality.
Security and Privacy Considerations for IoT Application on Smart Grids: Survey and Research Challenges	Management of vast amounts of data amongst others.	None	Consolidation of IoT challenges provided.	Study of data in transit.
The security challenges in the IoT enabled cyber-physical systems and opportunities for evolutionary computing & other computational intelligence	Attack surfaces indentified.	None	IoT architecture must be standardized. Design level equally important.	Areas of counter-threat measures identified.
IoT Security Development Framework for building trustworthy smart car services	Security is an afterthought in IoT.	Layered architecture and threat ranking protocols are prescribed.	False positives in sensor data are 4.2% of positives observed.	Blockchain architecture replacement considered.

**Table 3 sensors-23-00336-t003:** State Table: Behavior states of IoT insulin Pump.

Duration (h)	Meal Time	Insulin Status Expected	Ongoing Insulin Status	User Triggered	Inference
06:00 h to 08:00	✘	Basal	Basal	NA	No Deviation
		Basal	Bolus	Y	Update Care giver
		Basal	Bolus	N—Violation	Trigger Alert
08:01 h to 09:00	✓	Bolus	Bolus	NA	No Deviation
		Bolus	Basal	N—Violation	Trigger Alert
09:01 h to 12:30	✘	Basal	Basal	NA	No Deviation
		Basal	Bolus	Y	Update Care giver
		Basal	Bolus	N—Violation	Trigger Alert

**Table 4 sensors-23-00336-t004:** Premises considered for the Blockchain-IoT pump.

Statement	Axiom
Data in Blockchain is immutable.	BC
Two dosage settings basal and bolus. Basal and Bolus are mutually exclusive.	BS, BL; BS = !BL
Two triggers normal time and meal time. Normal time and meal time are mutually exclusive.	NT, MT; NT = !MT
Normal time triggers basal dosage state	NT→BS
Normal time basal dosage triggers continual loop.	BS→L
Meal time triggers bolus dosage.	MT→BL
Meal time bolus dosage triggers basal dosage at normal time.	BL^NT→BS
User can trigger bolus dosage.	U→ BL
Outsider should not change dosage.	!U→!BS; !U→!BL
Outsider changes dosage triggers loop break and triggers BC data entry & system halt.	(!U→BS)→BC^H(!U→BL)→BC^H
Registration and halt happens only during failure.	BC^H ←→!L

**Table 5 sensors-23-00336-t005:** Formal proof via contraposition and with malicious user intervention.

Proposition	Derivation
U→BL	Given
(!U→BS)→BC^H	Axiom (10)
!(BC^H)→!(!U→BS)	Contraposition (4)
!(BC^H)→(U→!BS)	Association (5)
!(BC^H)→(U→BL)	BS and BL are mutually exclusive
!(!L)→(U→BL)	Axiom (11)
L→(U→BL)	Double negation (8)
!U→!BL	Double negation (3)
!(U→BL)	Association (10)
!L	Modus Tollens of (9) & (11)
BC^H	Axiom (11)

## Data Availability

Not applicable.

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
