# Peer review of "Propositional Inference for IoT Based Dosage Calibration System Using Private Patient-Specific Prescription against Fatal Dosages"

_sensors, 2022, doi:10.3390/s23010336_

Round 1
Reviewer 1 Report
real-time deployment is not recommended by security experts. Why? reference.
The proposed approach establishes a patient-specific private Blockchain wherein.. Who authenticates the doctor and the nurses etc?
Can the nurses add blocks to the blockchain? if yes, then they can prescribe medicine. This is unethical.
Here, the proposed approach designates the doctor, chief doctor, nurse and caretaker of the patient as miners who are validating the addition of new blocks into the 287 Blockchain.. Validating a block is different from adding a block. There should be separation between nodes which add block, such as doctors, and those who validate, doctors, nurses, caretakers, etc..
for every patient there will be a separate private Blockchain is created thereby the unknown user participation is prohibited.. So if you have 1000 patients, you will have 1000 blockchains? this is impractical. distinguish between patient's records in the blockchain and the blockchain itself.
The instruction contained within the IoT pump will be the latest block containing the most recent prescription for the patient. The IoT device should be small compact and absolutely of limited capacity/memory. In other words, it is not supposed to store blocks. IoT do not have the capabilities to hold the entire blockchain. However, they listen for their address in the network. When they are being addressed, they trigger.
New dosage information block replaces... 2)the integrity of data contained within the IoT pump remains uncompromised. How? The integrity in intact since it is already on the blockchain. replacing the block in the IoT has to do with its memory only.
get the consensus from doctor, chief doctor, nurse and caretaker to add a new.. How would the attacker add a block if not in the system?
This can be circumvented by sending the block hash separately from the contents of the block.. As a matter of fact this increases the vulnerability. If you send the hash first, then the block, listening to these information repeatedly can compromise the keys of the senders.. Secondly, the point here is that the block is basically readable by all nodes in the chain. However, it can only be executed by the designated IoT.
gets the consensus from chief doctor, nurse and caretaker of the patient and appended.. There is difference between getting consensus and validating the block. The doctor generates the block which contains the prescription. Then appends its to the BC. Nodes then will validate.
Author Response
Dear Reviewer:
We would like to extend our gratitude for adding valuable constructive comments and suggestions on our submitted manuscript for its improvement. We have revised the manuscript in accordance with all suggestions, and detailed emendations are listed below.
Comment 1: real-time deployment is not recommended by security experts. Why? reference.
Response: The authors thank the anonymous reviewers for seeking this clarification. The real-time deployment involved insulin delivery to chronic diabetic patients and as such the proposed system requires approval from the Drugs Controller General of India (DCGI) and Central Drugs Standard Control Organization (CDSCO). Presently the proposed system showed success in a controlled environment which does not involve living subjects for testing. Thus real-time deployment requires multiple levels of compliance control test before real-time deployment.
Refer: https://morulaa.com/medical-device/step-by-step-process-to-register-your-medical-device-in-india/
Comment 2: The proposed approach establishes a patient-specific private Blockchain wherein.. Who authenticates the doctor and the nurses etc?
Response: The authors are thankful to the reviewer for pointing out this important observation. In order to lighten the strain on the system, the proposed prototype is premised on the belief that the doctor, chief doctor, nurse, and caretaker are noble agents who have been personally vetted by the hospice administration. At the moment, the main focus is on retrofitting IoT-Blockchain communication for seamless transfer. For system establishment, all forms for initial identity establishment are deemed to be complete.
Comment 3: Can the nurses add blocks to the blockchain? if yes, then they can prescribe medicine. This is unethical.
Response: The reviewer is correct in pointing out the ambiguity of block creating and appending regulation. The key personnel to create a block should be the presiding doctor for the patient for both implementation and ethical reasons. The descriptions corroborating the same have been reflected in the principal document/ paper. The corrections are highlighted in Red
Refer Pg: 8 Paragraph: 2: The proposed system consists of IoT based insulin pump and a patient-specific private Blockchain. The proposed IoT based insulin pump construction of patient-specific private Blockchain is as follows: The proposed approach establishes a patient-specific private Blockchain wherein the members need to get authenticated before part of the Blockchain thereby limiting the participation of unknown persons as a part of Blockchain to add and validate the blocks in the Blockchain.
Comment 4: Here, the proposed approach designates the doctor, chief doctor, nurse and caretaker of the patient as miners who are validating the addition of new blocks into the 287 Blockchain.. Validating a block is different from adding a block. There should be separation between nodes which add block, such as doctors, and those who validate, doctors, nurses, caretakers, etc..
Response: The authors thank the anonymous reviewer for highlighting the ambiguity. We agree with the reviewer on the need for clarity between addition and validation. The doctor is given the access to create & add a block in the chain while the rest of the miners collectively validate it via consent. The principal document reflects the change. The corrections are highlighted in Green.
Refer: Pg: 8 Paragraph: 2. The patient specific doctor is authorized to add blocks while the collective acknowledgement of chief doctor, nurse and care-taker validates the addition of new blocks into the Blockchain. In short, for every patient there will be a separate private Blockchain is created thereby the unknown user participation is prohibited.
Comment 5: for every patient there will be a separate private Blockchain is created thereby the unknown user participation is prohibited.. So if you have 1000 patients, you will have 1000 blockchains? this is impractical. Distinguish between patient's records in the blockchain and the blockchain itself.
Response: The authors thank the anonymous reviewers for seeking this clarification. We agree that creating one chain per patient is impractical and as such a solution for distinguishing patients in a chain containing multiple patients is proposed pending implementation for future study.
The corrections are highlighted in Purple.
Refer Page 8, Paragraph: 9: The limitations of the existing paradigm are that each patient can only have one new chain created for research. A blank header that separates patients with hash identification are taken into consideration when adding data from multiple patients in a single chain. One chain holds all the patient data; however it is restricted because it is an open data chain between credentialed medical professionals. For practical security reasons, it is advised that only one chain per authorized group of medical practitioners maintain their patients' information.
Comment 6: The instruction contained within the IoT pump will be the latest block containing the most recent prescription for the patient. The IoT device should be small compact and absolutely of limited capacity/memory. In other words, it is not supposed to store blocks. IoT do not have the capabilities to hold the entire blockchain. However, they listen for their address in the network. When they are being addressed, they trigger.
Response: The authors are thankful to the reviewer for pointing out this important observation. We agree that the IoT processor has very minimal and is incapable of storing the entire chain. But the present design requires the IoT device to hold the one block that contains prescription information. In said context, the augmented memory capacity of 2G via Micro SD card was sufficient to hold and verify block.
The corrections are highlighted in Aqua.
Refer: Page 8 Paragraph: 4.
In Bitcoin contexts, a block typically has a size of 1 MB and is constrained by a 4-byte field that limits the amount of data that may be stored in a block. Whereas, the current chain is required to store insulin prescription exclusively. Thus, reducing the minimum size of a block is not mandatory as the private Blockchain only communicates the dose information within the new prescription that is contained in the most recent block and not the entire chain, together with the hash content for IoT-pump validation. For the experiment, a 2 GB Micro SD / Memory Card and a 32 kB internal flash memory delivered the enhanced IoT-pump its optimum performance.
Comment 7: New dosage information block replaces... 2)the integrity of data contained within the IoT pump remains uncompromised. How? The integrity in intact since it is already on the blockchain. replacing the block in the IoT has to do with its memory only.
Response: The authors are thankful to the reviewer for pointing out this important observation. We agree with the findings of the reviewer. The data integrity is unaffected by replacement. The replacement however, ensures operational integrity of the IoT-pump which was ambiguously delivered by earlier statements. We have made changes in regards to this in the principal document to reflect the same. The authors apologize and request tolerance for the error.
The corrections are highlighted in White [Background 1, Darker ]
Refer Page 8 Paragraph 6: This is to ensure that the device is always using the recent dosage information which is in congruence with the most recent prognosis made by physician. This ensures the operational integrity of the IoT-Pump with strict diligence. . Hence, any attacker who attacks the IoT based insulin pump directly cannot modify the dosage information since it is available in the Blockchain
Comment 8: Get the consensus from doctor, chief doctor, nurse and caretaker to add a new.. How would the attacker add a block if not in the system?
Response: The authors thank the anonymous reviewers for seeking this clarification. The attacker cannot add any new blocks unless if he’s a miner and this is impossibility as all miners are declared initially. The statement was to emphasize robustness against subterfuge attacks which the inherent design of Blockchain-smart contracts are effective against.
The corrections are highlighted in Tan .
Refer Page 8 Paragraph 6: . This is impossible as the miners are declared openly, authenticated and fixed per patient as a clause of the smartcontract. Thus getting the consensus from doctor, chief doctor, nurse and caretaker to add a new block into the Blockchain is mitigated by inherent design of Private Blockchain Smartcontract. Ergo, there is no means for an attacker to see the contents of the private Blockchain unless he becomes a member through legitimate channels and screening process, ensuring patient safety
Comment 9: This can be circumvented by sending the block hash separately from the contents of the block.. As a matter of fact this increases the vulnerability. If you send the hash first, then the block, listening to these information repeatedly can compromise the keys of the senders.. Secondly, the point here is that the block is basically readable by all nodes in the chain. However, it can only be executed by the designated IoT.
Response: The authors are thankful to the reviewer for pointing out this important observation. The authors apologize for not addressing the potential threat to the system via separate transmissions and have made an experimental addendum via symmetric encryption to dampen identified vulnerability. The reflection is shown in the principal document and outlined below.
The corrections are highlighted in Green.
Refer Page 9 Paragraph 2: The other potentially, reasonable attack point occurs when the Blockchain and IoT insulin pump communicate dosage information. This can be avoided by providing the block hash, which is encrypted using the symmetric AES technique to increase security, independent of the block's content. To reduce processor burden, a key size of 128 bits was used. The process of introducing decryption required augmentation of Arduino processor with Raspberry Pi Pico/RP2040 with 8MB FLASH 264KB SRAM. This reduces the chance of listening attacks occurring while data is being transmitted. Despite adding to the stress on IoT processors, the decryption process serves as a firewall against passive attacks.
Comment 10: gets the consensus from chief doctor, nurse and caretaker of the patient and appended.. There is difference between getting consensus and validating the block. The doctor generates the block which contains the prescription. Then appends its to the BC. Nodes then will validate.
Response: The authors are thankful to the reviewer for pointing out this important observation. We agree with the findings of the reviewer. The ambiguity between consensus and validation has been refined that explicitly states that all miners with the exception of the prescription generating doctor will be required to validate the newly generated prescription block. We request the patience of the reviewer for the disparity.
The corrections are highlighted in Yellow.
Refer Page 12 Paragraph 1: The doctor decides the dosage information (i.e. specifies the quantity of insulin and the time of delivery) for the patient and generates the block in the chain containing latest prescription information. The block is validated by the consensus mechanism that includes chief doctor, nurse and caretaker of the patient and appended into the Blockchain.

Reviewer 2 Report
The following should be noted and corrected accordingly:
1. How practicable is your proposed model in real-time?
2. Is it cost-efficient?
3. Some diagrams and terms are not properly explained.
4. Grammar requires minor re-editing
5. Are the formulas and numbers here generic or generated by you?
6. What is the future scope of the study?
7. Study and consider the following related paper to embellish your paper:
https://doi.org/10.1155/2022/9077348
Author Response
Comment 1: How practicable is your proposed model in real-time?
Response: The authors thank the anonymous reviewers for seeking this clarification. The proposed system was able to automate the drug dispensary cannula in a simulated environment for a continuous dry run of 45 minutes. It responded properly to the incoming data. The delay due to transmission by the Blockchain addendum needs study. In the 45 minute duration of study a total of 7 individual transactions were documented at a bandwidth of 150 mbps in a public network.
Comment 2: Is it cost-efficient?
Response: The authors thank the anonymous reviewers for seeking this clarification. The cost of a traditional insulin pump is a 1000$ U.S (approx). The proposed augmentation via Arduino based micro controllers with Blockchain remote oversight is around 100$ U.S. Thus for an additional tenth of the traditional pumps cost, the medical actuator is made secure.
Comment 3: Some diagrams and terms are not properly explained.
Response: We have noted the reviewer’s concern on the images in our original manuscript. As per the suggestion, the revised manuscript has now contains all figures with explanations accompanying them. The diagrams have been reproduced in high quality (150 dpi).
Comment 4: Grammar requires minor re-editing
Response: The authors are thankful to the reviewer for his suggestion to improve the English language. As per the suggestion, the flaws pointed out have been rectified and the revised manuscript has been thoroughly proofread for any such typos and grammatical errors.
Comment 5: Are the formulas and numbers here generic or generated by you?
Response: The authors thank the anonymous reviewers for seeking this clarification. The formulae, numbers and facts are generated by the authors based on propositional premises. These premises are derived from study of the instrument, problem statements identified in various references and validated by UPPAAL model checking software.
Comment 6: What is the future scope of the study?
Response: Future scope of applications includes retrofitting similar pumps to automate intravenous drug delivery mechanisms to ensure accountability in all aspects of medical intervention.
Comment 7: Study and consider the following related paper to embellish your paper:
https://doi.org/10.1155/2022/9077348
Response: The authors thank the reviewer for their suggestion. We have studied the paper and have incorporated the papers facts to augment the principle papers statements on Blockchain and its applications in arenas other than digital currency.

Reviewer 3 Report
This study propose system based on Blockchain methodology to protect IoT-based Insulin pumps from hardware and software attacks. Although the method could be promising, the quality of the overall presentation is not satisfactory for publication.
The most significant drawbacks are:
1. Abstract should be rewritten. The abstract should be mainly oriented toward describing the proposal of the new system and the results obtained.
2. References are not listed in the proper order. In the introduction, the first reference mentioned is number 11.
3. Ln 84-86 the paper's main contributions are not stated.
4. Figures 4.1., and 4.2. The quality of figures and fonts should be improved.
5. Ln 399: Avoid section titles ending with “:”
Author Response
Comment 1: Abstract should be rewritten. The abstract should be mainly oriented toward describing the proposal of the new system and the results obtained.
Response: The authors are thankful to the reviewer for pointing out this important observation. We have noted the reviewer’s concern on it. As per the reviewer’s suggestion, the abstract section have been revised and these sections have been written in blue for easy identification.
Comment 2: References are not listed in the proper order. In the introduction, the first reference mentioned is number 11.
Response: The authors are thankful to the reviewer for pointing out this important observation. The authors apologize for disorganized citations and corresponding reference ordering. As per the reviewer’s suggestion, the citations sections have been revised and the references have been ordered in an ascending manner.
Comment 3: Ln 84-86 the paper's main contributions are not stated.
Response: The authors are thankful to the reviewer for pointing out this important observation. The authors apologize for the typographic error incurred. The error is deleted in the principal document.
Comment 4: Figures 4.1., and 4.2. The quality of figures and fonts should be improved.
Response: We have noted the reviewer’s concern on the quality of the images and its fonts. As per the suggestion, the revised manuscript has now contains all figuresthat have been reproduced in high quality (600 dpi).
Comment 5:Ln 399: Avoid section titles ending with “:”
Response: We have noted the reviewer’s concern on blurry images in our original manuscript. As per the suggestion, all the sections in the revised manuscript has been verified to implement said suggestion.

Round 2
Reviewer 1 Report
There is no need to save the block at all. The blocks are written on the blockchain which will remain forever and it will be remain accessible.
Response: The authors are thankful to the reviewer for pointing out this important observation. We agree that the IoT processor has very minimal and is incapable of storing the entire chain. But the present design requires the IoT device to hold the one block that contains prescription information. In said context, the augmented memory capacity of 2G via Micro SD card was sufficient to hold and verify block.
Reviewer 2 Report
No further recommendations